# JMJD6 Autoantibodies as a Potential Biomarker for Inflammation-Related Diseases

**DOI:** 10.3390/ijms25094935

**Published:** 2024-04-30

**Authors:** Bo-Shi Zhang, Xiao-Meng Zhang, Masaaki Ito, Satoshi Yajima, Kimihiko Yoshida, Mikiko Ohno, Eiichiro Nishi, Hao Wang, Shu-Yang Li, Masaaki Kubota, Yoichi Yoshida, Tomoo Matsutani, Seiichiro Mine, Toshio Machida, Minoru Takemoto, Hiroki Yamagata, Aiko Hayashi, Koutaro Yokote, Yoshio Kobayashi, Hirotaka Takizawa, Hideyuki Kuroda, Hideaki Shimada, Yasuo Iwadate, Takaki Hiwasa

**Affiliations:** 1Department of Neurological Surgery, Graduate School of Medicine, Chiba University, Chiba 260-8670, Japan; dr.boshizhang@gmail.com (B.-S.Z.);; 2Department of Biochemistry and Genetics, Graduate School of Medicine, Chiba University, Chiba 260-8670, Japan; 3Department of Clinical Oncology, Graduate School of Medicine, Toho University, Tokyo 143-8541, Japanhideaki.shimada@med.toho-u.ac.jp (H.S.); 4Department of Gastroenterological Surgery, Graduate School of Medicine, Toho University, Tokyo 143-8541, Japan; 5Department of Cardiovascular Medicine, Graduate School of Medicine, Kyoto University, Kyoto 606-8507, Japan; 6Department of Pharmacology, Shiga University of Medical Science, Otsu 520-2192, Japan; 7Comprehensive Stroke Center, Chiba University Hospital, Chiba 260-8677, Japan; 8Department of Neurological Surgery, Chiba Prefectural Sawara Hospital, Chiba 287-0003, Japan; 9Department of Neurological Surgery, Chiba Cerebral and Cardiovascular Center, Chiba 290-0512, Japan; 10Department of Neurosurgery, Eastern Chiba Medical Center, Chiba 283-8686, Japan; 11Department of Endocrinology, Hematology and Gerontology, Graduate School of Medicine, Chiba University, Chiba 260-8670, Japan; 12Department of Diabetes, Metabolism and Endocrinology, School of Medicine, International University of Health and Welfare, Chiba 286-8686, Japan; 13Department of Cardiovascular Medicine, Graduate School of Medicine, Chiba University, Chiba 260-8670, Japan; 14Port Square Kashiwado Clinic, Kashiwado Memorial Foundation, Chiba 260-0025, Japan; 15Medical Project Division, Research Development Center, Fujikura Kasei Co., Saitama 340-0203, Japan

**Keywords:** JMJD6, inflammation, ischemic stroke, acute myocardial infarction, diabetes mellitus, atherosclerosis, esophageal cancer, autoantibody

## Abstract

Inflammation is closely associated with cerebrovascular diseases, cardiovascular diseases, diabetes, and cancers, and it is accompanied by the development of autoantibodies in the early stage of inflammation-related diseases. Hence, it is meaningful to discover novel antibody biomarkers targeting inflammation-related diseases. In this study, Jumonji C-domain-containing 6 (JMJD6) was identified by the serological identification of antigens through recombinant cDNA expression cloning. In particular, JMJD6 is an antigen recognized in serum IgG from patients with unstable angina pectoris (a cardiovascular disease). Then, the serum antibody levels were examined using an amplified luminescent proximity homogeneous assay-linked immunosorbent assay and a purified recombinant JMJD6 protein as an antigen. We observed elevated levels of serum anti-JMJD6 antibodies (s-JMJD6-Abs) in patients with inflammation-related diseases such as ischemic stroke, acute myocardial infarction (AMI), diabetes mellitus (DM), and cancers (including esophageal cancer, EC; gastric cancer; lung cancer; and mammary cancer), compared with the levels in healthy donors. The s-JMJD6-Ab levels were closely associated with some inflammation indicators, such as C-reactive protein and intima–media thickness (an atherosclerosis index). A better postoperative survival status of patients with EC was observed in the JMJD6-Ab-positive group than in the negative group. An immunohistochemical analysis showed that JMJD6 was highly expressed in the inflamed mucosa of esophageal tissues, esophageal carcinoma tissues, and atherosclerotic plaques. Hence, JMJD6 autoantibodies may reflect inflammation, thereby serving as a potential biomarker for diagnosing specific inflammation-related diseases, including stroke, AMI, DM, and cancers, and for prediction of the prognosis in patients with EC.

## 1. Introduction

Inflammation is a primary pathological process in atherosclerosis [1,2,3], playing a key role in the pathogenesis of diabetes mellitus (DM) [2,3], and is also associated with various cancers [4,5]. Atherosclerosis, which is typically accompanied by inflammation, is a leading risk factor for vascular diseases, including acute cerebral infarction (ACI) and acute myocardial infarction (AMI) [6,7,8]. In addition, inflammatory responses can lead to insulin resistance, a key characteristic of type 2 diabetes [9]. Moreover, prolonged inflammatory states can cause sustained damage and cellular stimulation, thereby increasing the risk of cancer occurrence [10,11]; for example, due to the tumor-promoting effects of chronic intestinal inflammation, patients with inflammatory bowel disease are at a higher risk of developing colorectal cancer [12]. Therefore, exploring novel biomarkers for inflammation-related diseases not only sheds light on the complex interplay between inflammation and various pathological conditions but also holds promise for the development of novel diagnostic and therapeutic strategies, thereby advancing personalized medicine in the management of vascular diseases, DM, and cancers.

Autoantibodies are antibodies produced by the immune system against the body’s own proteins or antigens. Normally, the immune system recognizes self-antigens as “self” and does not mount an immune response against them. However, if the immune system recognizes self-antigens, it starts to produce autoantibodies, which can contribute to tissue damage [13,14]. Autoantibodies have been implicated in autoimmune diseases, cancers, atherosclerosis, and DM, and they can also serve as biomarkers for disease diagnosis [13,14,15,16]. Previously, we have searched for antibody markers using serological identification through cDNA expression cloning (SEREX) and protein array methods. Our findings highlighted potential biomarkers for stroke, including DIDO1, CPSF2, FOXJ2, and TSTD2 [17,18]. Similarly, we identified striatin 4 as a potential marker for digestive tract cancers [19].

In the present study, Jumonji C-domain-containing 6 (JMJD6) was identified through SEREX using serum from a patient with unstable angina pectoris (UAP), a cardiovascular disease. JMJD6 is a member of the Jumonji C-domain-containing family and has emerged as a promising target for various diseases, including breast cancer, prostate cancer, and Alzheimer’s disease [20,21,22,23]. JMJD6 functions as a protein demethylase towards histone substrates, a hydroxylase targeting non-histone proteins, and a phosphatidylserine receptor influencing RNA splicing, lipid metabolism, and apoptosis [22,23,24,25]. Considering its diverse functions, JMJD6 may be involved in the development and progression of various diseases, and it shows the potential to serve as a target for the diagnosis and treatment of multiple diseases. Despite ample evidence supporting the pivotal role of JMJD6 in the treatment and prognosis of patients with cancer [22,25] and its relevance in managing ischemic cardiovascular diseases [26], a noticeable lack of research is observed regarding the autoantibodies against JMJD6. In this study, we aimed to explore the potential role of serum anti-JMJD6 antibodies (s-JMJD6-Abs) in inflammatory-related diseases and to consider them as biomarkers for inflammatory myocardial infarction, stroke, DM, and cancers. While it is understood that there is a close association between these diseases and inflammatory responses, the discovery of s-JMJD6-Abs as biomarkers can still aid in early diagnosis, treatment, and prognosis, providing crucial clinical guidance for medical practice.

## 2. Results

### 2.1. Recognition of JMJD6 Using Serum Antibodies from Patients with UAP

This study comprehensively screened antigens recognized by SEREX using serum IgG antibodies from patients with UAP, which is considered to be an early stage of AMI. Furthermore, one of the antigens identified was the 1075–1536 region of the *JMJD6* gene (accession no. NM_015167). To further investigate its potential as a disease marker, we expressed the glutathione S-transferase (GST)-fused JMJD6 protein in *Escherichia coli* (*E. coli*) and purified it using affinity chromatography. The purified protein was subsequently utilized as an antigen to evaluate the serum antibody levels.

### 2.2. Elevation of the Levels of Serum Antibodies against JMJD6 in Patients with Cerebrovascular Diseases

We conducted an amplified luminescence proximity homogeneous assay-linked immunosorbent assay (AlphaLISA) and investigated the levels of s-JMJD6-Abs in healthy donors (HDs) and patients with chronic cerebral infarction (CCI), ACI, transient ischemic attack (TIA), asymptomatic cerebral infarction (asympt-CI), or deep and subcortical white matter hyperintensity (DSWMH; Table 1, upper panel). The patients with TIA or asympt-CI were grouped together due to their similar characteristics. s-JMJD6-Ab levels significantly increased in patients with CCI, ACI, TIA/Asympt-CI, and DSWMH, compared with those in HDs (Figure 1a). When we employed a cutoff value equivalent to the 95% percentile value of HDs, the positive rates of s-JMJD6-Abs in HDs and patients with CCI, ACI, TIA/Asympt-CI, and DSWMH were 4.9%, 20.0%, 21.8%, 15.3%, and 8.6%, respectively (Table 1, lower panel). The ability of s-JMJD6-Ab markers to indicate the presence of CCI, ACI, TIA/Asympt-CI, and DSWMH was evaluated through receiver operating characteristic (ROC) analysis, and the results indicated that the area under the ROC curve (AUC) values for s-JMJD6-Abs were 0.691, 0.708, 0.692, and 0.594, respectively (Figure 1b–e).

### 2.3. Analysis of Correlations between JMJD6 Antibody Levels and Clinical Parameters in the Sawara Hospital Cohort

To explore the potential correlations between s-JMJD6-Ab levels and clinical parameters in the Sawara Hospital cohort, we compared the s-JMJD6-Ab levels among participants with different characteristics using the Mann–Whitney *U* test. Specifically, we analyzed the s-JMJD6-Ab levels between males and females, body mass index (BMI) < 25 kg/m^2^ and BMI ≥ 25 kg/m^2^, and the presence and absence of DM, HT, cardiovascular diseases (CVDs), lipidemia, habitual smoking, and alcohol intake (Figure 1f–m). The s-JMJD6-Ab levels were significantly higher in those with DM, HT, and CVDs than in those without these diseases (*p* < 0.01, *p* < 0.0001, and *p* < 0.01, respectively; Figure 1h–j). In addition, smokers exhibited increased levels of s-JMJD6-Abs compared with non-smokers (*p* < 0.001, Figure 1i); notably, smoking is a significant factor for vascular diseases [27]. However, other categories showed no significant difference in s-JMJD6-Ab levels (Figure 1f,g,k,m). Taken together, the s-JMJD6-Ab levels may be associated with DM, HT, and CVDs, as well as smoking habit, in the Sawara Hospital cohort.

Moreover, the relationships between the s-JMJD6-Ab levels and various clinical variables in the Sawara Hospital cohort were investigated using Spearman’s correlation analysis. Table 2 presents the correlation coefficients (Rho) and *p*-values for patients’ hematological examination results, smoking status, alcohol consumption status, age, sex, BMI, and intima–media thickness (IMT) of the carotid artery. The s-JMJD6-Ab levels were positively correlated with maximum IMT (Rho = 0.2761, *p* < 0.0001), alkaline phosphatase (Rho = 0.0989, *p* = 0.0059), C-reactive protein (Rho = 0.1370, *p* = 0.0007), white blood cell count (WBC) (Rho = 0.0702, *p* = 0.0426), neutrophils (Rho = 0.0818, *p* = 0.0331), blood glucose (Rho = 0.1343, *p* = 0.0002), blood pressure (Rho = 0.0946, *p* = 0.0238), and smoking duration (Rho = 0.1515, *p* < 0.0001). In contrast, the s-JMJD6-Ab levels were negatively correlated with total cholesterol (Rho = −0.1239, *p* = 0.0008), triglyceride (Rho = −0.0989, *p* = 0.0173), and lymphocytes (Rho = −0.1083, *p* = 0.0047) (Table 2). Overall, the s-JMJD6-Ab levels may be correlated with inflammation-associated factors such as C-reactive protein and WBC, as well as certain vascular risk factors such as IMT and blood pressure. Therefore, the s-JMJD6-Ab levels are potentially linked to inflammation and vascular diseases in the Sawara Hospital cohort. However, s-JMJD6-Abs exhibited no correlation with eosinophils and monocytes, which are associated with infectious diseases and inflammation [28]. Consequently, s-JMJD6-Abs may not simply manifest in all inflammatory contexts but may be indicative of specific inflammation-related diseases such as atherosclerosis and/or ischemic stroke.

Using the clinical risk factors and cutoff value of 9274 based on the ROC curve analysis for the ACI group, which was applied in the univariate analysis, the logistic regression analysis in the ACI group revealed that not only clinical risk factors such as age over 65 years (*p* < 0.0001), DM (*p* < 0.0001), HT (*p* < 0.0001), and CVDs including AMI and UAP (*p* = 0.0026) but also high antibody levels (*p* < 0.0001) were associated with ACI (Table 3). Multivariate analysis of the univariate data with *p*-values of <0.05 revealed that, in addition to the atherosclerotic factors mentioned earlier, elevated s-JMJD6-Ab levels were an independent predictor of ACI (odds ratio: 3.77, 95% CI: 2.19–6.50, *p* < 0.0001; Table 3). These results suggested that s-JMJD6-Ab levels may serve as an independent biomarker for ACI, helping to improve the prediction and diagnosis of ACI risk.

### 2.4. Elevation of s-JMJD6-Ab Levels in Patients with AMI and DM

Using AlphaLISA, we examined the s-JMJD6-Ab levels in age-matched HDs and patients with AMI and DM (Table 4, upper panel). The s-JMJD6-Ab levels were significantly higher in patients with AMI and DM than in HDs (Figure 2a). Using the cutoff value for JMJD6-Ab levels defined at the 95% percentile of the s-JMJD6-Ab level in HDs, the positivity rates of s-JMJD6-Abs were 21.9% in patients with AMI and 21.1% in patients with DM (Table 4, lower panel). The ROC analysis revealed AUC values of 0.716 for patients with AMI and 0.661 for patients with DM (Figure 2b,c), indicating a strong association of s-JMJD6-Ab levels with AMI and DM.

### 2.5. Elevated s-JMJD6-Ab Levels in Patients with Cancers

Considering previous reports on the correlation between JMJD6 and cancers, as well as the link between inflammation and cancers, we conducted further investigations using serum samples from patients with esophageal cancer (EC), gastric cancer (GC), lung cancer (LC), and mammary cancer (MC) (Table 5, upper panel). The s-JMJD6-Ab levels were significantly higher in these patients compared with those in HDs (Figure 3a and Table 5, lower panel). The positive rates of JMJD6-Ab were high in patients with EC and GC (34.9% and 30.2%, respectively; Table 5, lower panel). In addition, the AUCs of s-JMJD6-Abs were 0.721, 0.707, 0.585, and 0.588 for EC, GC, LC, and MC, respectively (Figure 3b–e). The s-JMJD6-Ab levels may be more closely associated with EC and GC than other cancer types. Thus, s-JMJD6-Ab levels could serve as useful biomarkers for detecting EC, GC, LC, and MC, with a particular emphasis on EC and GC.

Subsequently, we then investigated the association between s-JMJD6-Ab levels and survival outcomes in patients with EC and GC using retrospectively collected survival data. Patients with EC and GC were categorized into positive and negative groups based on the cutoff values obtained from the X-tile software (Version 3.6.1). In patients with EC, the overall survival status of the s-JMJD6-Ab-positive group was significantly better than that of the negative group (Figure 3f). Conversely, no significant difference was noted between the s-JMJD6-Ab-positive and -negative groups in patients with GC (Figure 3g). Therefore, JMJD6-Ab levels can predict the prognosis in patients with EC but not with GC.

### 2.6. Combinatorial Prognosis Prediction Combining the s-JMJD6-Ab Levels with Programmed Cell Death Ligand 1 (PD-L1)

Given that serum PD-L1 levels were also reported to be associated with EC prognosis [29], we examined the prognostic effects of PD-L1 alone or in combination with the s-JMJD6-Abs in EC specimens. We established a cutoff value of PD-L1 to obtain the lowest *p*-value in survival analysis using the X-tile software (Version 3.6.1). The prognosis of the PD-L1-positive group was significantly more unfavorable than that of the PD-L1-negative group (*p* = 0.0007, Figure 3h). Moreover, the prognosis differed more significantly between the s-JMJD6-Ab-negative/PD-L1-positive group and the s-JMJD6-Ab-positive/PD-L1-negative group (*p* < 0.0001, Figure 3i). Thus, JMJD6 antibodies may have a potential association with PD-L1 expression, highlighting the significance of their interaction with patient survival.

In addition, we performed prognosis analysis using JMJD6 mRNA expression levels collected from The Cancer Genome Atlas (TCGA) database in OncoLnc, and the result shows that patients with high JMJD6 expression have worse prognosis than those with low JMJD6 expression (*p* = 0.0293; Figure 3j). JMJD6 mRNA expression has a different impact on survival outcomes in patients with EC compared to the effect of s-JMJD6-Abs (Figure 3f,j). This may suggest an antagonistic relationship between the JMJD6 protein and the autoantibodies.

### 2.7. Immunohistochemical Analysis of JMJD6

Assuming that autoantibodies against JMJD6 can develop in patients with inflammation-related diseases, JMJD6 should be expressed at high levels in inflamed lesions. Thus, we examined the expression of the JMJD6 protein in a surgically resected normal esophagus, the inflamed mucosa of the esophagus, esophageal carcinoma, and atherosclerotic plaques via immunohistochemistry. Our findings demonstrate high JMJD6 expression in inflamed esophagus tissue samples and esophageal carcinoma, but not in normal esophagus mucosa (Figure 4a–c). JMJD6 was highly expressed in smooth muscle cells (SMCs) in atherosclerotic plaques, according to the same localization with SMC markers, vimentin (VIM) and smooth muscle actin (SMA) (Figure 4d–f), and previous reports [17]. These results indicate JMJD6 is highly expressed in inflamed tissues, underscoring the potential of JMJD6 as a marker for inflammation.

## 3. Discussion

This study investigated the relationship between the levels of s-JMJD6-Abs and various inflammation-related diseases, including ischemic stroke, DM, AMI, and cancers. First, the levels of s-JMJD6-Abs in patients with CCI, ACI, AMI, DM, EC, GC, LC, and MC were considerably higher than those in HDs (Figure 1a, Figure 2a and Figure 3a). TIA, asympt-CI, and DSWMH are recognized as risk factors and/or early symptoms for future stroke [30,31]. The levels of s-JMJD6-Abs showed a slight but significant elevation in patients with TIA/Asympt-CI and DSWMH, suggesting a potential association with the early stages of ischemic stroke. This may be because JMJD6 can function in the progress of inflammation (Table 2), and atherosclerosis accompanied by inflammation is a major cause of cerebral infarction [1,32]. Therefore, s-JMJD6-Abs can be used to predict patients with cerebral infarction risk at the early stage.

The Sawara Hospital cohort showed that s-JMJD6-Ab levels were strongly correlated with IMT of the carotid artery (Table 2). This also suggests that s-JMJD6-Abs primarily reflect the progression of atherosclerosis, which may lead to the onset of ischemic stroke and AMI. Moreover, s-JMJD6-Abs were positively correlated with neutrophils but not eosinophils, basophils, monocytes, and lymphocytes (Table 2). Although neutrophils, eosinophils, basophils, monocytes, and lymphocytes fall under the category of WBC, their sensitivity and significance vary for different diseases. For instance, neutrophils have traditionally been viewed as bystanders or biomarkers of vascular diseases [33]. These results indicate the role of s-JMJD6-Abs in some specific inflammation-associated diseases such as cerebrovascular diseases. When comparing two groups of clinical parameters using the Mann–Whitney *U* test, s-JMJD6-Ab levels were significantly higher in subjects with HT or a history of habitual smoking compared with those without these risk factors (Figure 1i,l). HT and smoking are typical risk factors not only for vascular diseases, but also for cancers [34,35,36,37], which aligns with the elevated s-JMJD6-Ab levels in patients with cancers (Figure 3a). In addition, individuals with DM or CVDs showed higher s-JMJD6-Ab levels than those without these conditions in the Sawara Hospital cohort (Figure 1h,j), consistent with the results in Figure 2a, which compares HDs with patients with AMI and DM.

Immunohistochemical staining showed that JMJD6 was highly expressed in SMCs, and colocalized with SMC markers such as VIM and SMA (Figure 4d–f) [17] in SMCs. SMCs actively contribute to inflammation in the lesion because of their acquired capacity to produce inflammatory mediators [38,39]. Hence, JMJD6 may also contribute to the inflammatory process in atherosclerosis. Furthermore, inflammation persists throughout the stages of atherosclerosis, contributing to plaque instability and promoting the cycle of damage and repair [2,40]. CRP and WBC, as important indicators of inflammation, are positively correlated with JMJD6 (Table 2). The inflamed tissues showed high levels of JMJD6 expression, compared with the normal tissues (Figure 4). This further supports the notion that s-JMJD6-Abs may predict the occurrence of atherosclerosis through their inflammatory effects, thus exhibiting higher levels in the sera of patients with cerebrovascular or cardiovascular diseases. According to the multivariate analysis, s-JMJD6-Abs emerged as an independent predictor in ACI (Table 3). These findings underscore the association between s-JMJD6-Abs and the development of atherosclerosis by inflammation, culminating in the onset of ACI. 

The increase in autoantibodies is primarily attributed to cellular overexpression and subsequent tissue damage, which leads to the release of antigen proteins into the bloodstream, triggering an immune response [41,42]. For instance, the antigenic proteins of other atherosclerosis autoantibody markers, DIDO1-Abs, CPSF2-Abs, and FOXJ2-Abs, were highly expressed in the carotid atherosclerotic plaques [17]. Similarly, cancer autoantibody biomarkers, such as SH3GL1-Abs and striatin 4-Abs, were associated with high antigen expression in the cancer tissues [19,43]. Thus, the elevated levels of s-JMJD6-Abs may be caused by increased JMJD6 expression in patients with inflammation-related diseases. So, what is the role of JMJD6? It can promote apoptosis through protein demethylation and hydroxylation in cancers and vascular diseases [44,45], which might explain its role in inflammation.

Next, we analyzed the relationship between s-JMJD6-Abs and other biomarkers in the prognosis of patients with EC. Our survival analysis showed the favorable prognosis of patients with EC in the s-JMJD6-Ab-positive group compared to in the s-JMJD6-Ab-negative group (Figure 3f). However, patients with high JMJD6 mRNA expression showed decreased overall survival rates compared to those with low expression in patients with EC (Figure 3j). This phenomenon may be due to the ability of JMJD6 to regulate the immune system through promoting the apoptosis of immune cells and influencing the corresponding inflammatory process [45,46]; furthermore, JMJD6 can facilitate apoptosis of the immune cells, aiding tumor immune evasion and leading to unfavorable prognosis for cancer patients. Higher levels of JMJD6-Abs can inhibit the apoptosis function of JMJD6, resulting in a better prognosis.

The combined analysis of s-JMJD6-Abs and PD-L1 in patients with EC suggests that JMJD6 may be functionally related to PD-L1 in EC cells. Cancer cells may escape and spread due to the inhibition of cell apoptosis by PD-L1 [42,47,48]. In contrast, JMJD6 is a phosphatidylserine receptor, and a monoclonal anti-JMJD6 antibody inhibited the phagocytosis of phosphatidylserine-expressing apoptotic cells [24,49]. Studies have shown that the knockout of the JMJD6 gene results in a larger number of non-phagocytosed apoptotic cells compared to wild-type animals [50,51], inhibiting the phagocytosis of phosphatidylserine-expressing apoptotic cells [24,49]. Thus, a similar phenomenon could also be caused by s-JMJD6-Abs. JMJD6-Abs and PD-L1 have opposite effects on apoptosis, which could be reflected in their levels with respect to prognosis. Additionally, Spearman’s correlation analysis showed that there was no correlation between PD-L1 and s-JMJD6-Ab levels (Appendix A), which might be because apoptosis can be influenced by many other biomarkers. This may account for the more accurate prediction of EC prognosis with a combination of PD-L1 and s-JMJD6-Abs than either alone.

Previous research has suggested the association between JMJD6 and many diseases, especially s-JMJD6-Ab and colorectal cancer [52]. Our study identifies a potential association between s-JMJD6-Abs and various inflammation-related diseases, and it suggests JMJD6-Abs as a biomarker for inflammation-related diseases. However, we acknowledge the limitations regarding the predictive performance of s-JMJD6-Abs alone, as indicated by the modest AUC values. Using AlphaLISA, we confirmed that s-JMJD6-Abs can serve as a potential biomarker for inflammation-related diseases including cerebrovascular diseases, cardiovascular diseases, DM, and cancers. However, due to the technical limitations of AlphaLISA, the output data may not be consistently stable. Therefore, for its practical application, larger-scale studies and more accurate research methods are needed to determine cutoff values for JMJD6-Abs under different conditions. In addition, there are already some biomarkers for atherosclerosis, such as serum interleukin-1β, interleukin-6, and homocysteine levels [53], and the mechanisms of the effect of JMJD6 and other biomarkers of inflammation differ. Therefore, further studies are necessary to elucidate the intrinsic connections between different biomarkers and the broader effects of JMJD6 autoantibodies under different pathological conditions.

## 4. Materials and Methods

### 4.1. Patient and Control Sera

Serum samples were collected from HDs and patients with CCI, ACI, TIA, Asympt-CI, and DSWMH from Chiba Prefectural Sawara Hospital. The stroke subtypes were determined according to the criteria of the Trial of Org 10,172 in the Acute Stroke Treatment classification system [54], and ACI includes large-artery atherosclerosis and small-artery occlusion (lacune). For comparisons with TIA and ACI, serum samples from HDs were obtained from patients without abnormalities in cranial magnetic resonance imaging. In addition, the sera of patients with AMI were obtained from Kyoto University Hospital, and those of patients with UAP or DM were collected from Chiba University Hospital. The serum samples of patients with TIA, ACI, AMI, and UAP were obtained within 2 weeks after disease onset. The Department of Surgery in Toho University Hospital collected sera from patients with EC, GC, LC, and MC preoperatively between June 2010 and February 2016, and patients with EC or GC were retrospectively followed up until July 2018 or death.

All sera were collected from patients with written informed consent. Each serum sample was centrifuged at 3000× *g* for 10 min, and the supernatant was stored at −80 °C before use to avoid the repeated freezing/thawing of samples. This study was approved by the Local Ethical Review Board of the Chiba University, Graduate School of Medicine (approval no.: 2018-320), and the Ethics Committee of Toho University Graduate School of Medicine (nos. A18103_A17052_A16035_A16001_26095_25024_24038_22047, M21038_20197_19213), as well as the review boards of the participating hospitals.

### 4.2. SEREX Screening of the Expressed Recombinant Proteins Using a Human cDNA Library

We utilized a modified method for immunoscreening. To identify clones that displayed immunoreactivity against the serum IgG antibodies in patients with UAP, we employed a human aortic endothelial cell cDNA library from the Uni-ZAP XR Premade Library (Stratagene, La Jolla, CA, USA). Then, we infected *E. coli* XL1-Blue MRF′ with Uni-ZAP XR phage and induced the expression of resident cDNAs by blotting infected bacteria onto nitrocellulose membranes (NitroBind, Osmonics, Minnetonka, MN, USA) pretreated with 10 mM isopropyl-β-D-thiogalactopyranoside (IPTG) (Wako Pure Chemicals, Osaka, Japan) for 30 min.

Next, we washed the membranes with TBS-T (20 mM Tris–HCl (pH 7.5), 0.15 M NaCl, and 0.05% Tween-20) and blocked non-specific binding by incubating them with 1% protease-free bovine serum albumin (Nacalai Tesque, Inc., Kyoto, Japan) in TBS-T for 1 h. The membranes were then incubated overnight with 1:2000 diluted sera. After three washes with TBS-T, these membranes were treated for 1 h with 1:5000 diluted alkaline phosphatase-conjugated goat antihuman IgG (Jackson ImmunoReseach Laboratories, West Grove, PA, USA).

When incubating the membranes in a color development solution (100 mM Tris–HCl (pH 9.5), 100 mM NaCl, and 5 mM MgCl_2_), which contained 0.15 mg/mL 5-bromo-4-chloro-3-indolylphospate (Wako Pure Chemicals) and 0.3 mg/mL nitro blue tetrazolium (Wako Pure Chemicals), we detected positive reactions. To obtain monoclonality, we re-cloned positive clones twice further, as described.

### 4.3. Sequence Analysis of Isolated cDNAs

To convert monoclonalized phage cDNA clones, we utilized ExAssist helper phage (Stratagene) for excision, resulting in pBluescript phagemid formation. The plasmid DNA was obtained from the *E. coli* SOLR strains that had been transformed with the phagemids. After sequencing the inserted cDNAs, we conducted homologous analysis using a public database provided by the National Center for Biotechnology Information (accessed on 9 January 2024, https://blast.ncbi.nlm.nih.gov/Blast.cgi).

### 4.4. Expression and Purification of JMJD6 Protein

To generate the expression plasmids for the GST-fused JMJD6 protein, we integrated the JMJD6 cDNA sequence between 1075 and 1834 into the EcoRI/XhoI site of pGEX-4T-1 vector (Cytiva, Marlborough, MA, USA), as previously described [17].

Moreover, *E. coli* BL-21 cells transformed with the pGEX-4T-1 clone were cultured in 200 mL of Luria–Bertani broth and treated with 0.1 mM IPTG for 3 h. The cells were then collected and lysed by sonication in BugBuster Master Mix (Novagen, San Diego, CA, USA), followed by centrifugation at 13,000× *g* and 4 °C for 10 min. GST-tagged JMJD6 proteins were purified through Glutathione-Sepharose column chromatography (GE Healthcare Life Sciences, Chicago, IL, USA) and dialyzed as previously described [17,18,19].

### 4.5. Amplified Luminescence Proximity Homogeneous Assay-Linked Immunosorbent Assay

AlphaLISA, a commercial technology that can detect target molecules in serum and plasma, was conducted in 384-well microtiter plates (white opaque OptiPlate™, Revvity, Waltham, MA, USA) containing 2.5 µL of 1:100 diluted serum with either 2.5 µL of GST or GST-JMJD6 proteins (10 µg/mL) in AlphaLISA buffer (25 mM N-2-hydroxyethylpiperazine-N-2-ethane sulfonic acid, pH 7.4, 0.1% casein, 0.5% Triton X-100, 1 mg/mL dextran-500, and 0.05% ProClin-300). The serum samples collected from HDs and patients diagnosed with CCI, ACI, TIA, Asympt-CI, and DSWMH at Chiba Prefectural Sawara Hospital were examined simultaneously. Serum autoantibodies against JMJD6 from patients with AMI, DM, and HDs serving as controls were also used concurrently. Serum samples from cancer patients and their corresponding HD controls were simultaneously utilized to measure the s-JMJD6-Abs. The reaction mixture was incubated at room temperature for 6–8 h and added with anti-human IgG-conjugated acceptor beads (2.5 µL at 40 µg/mL) and glutathione-conjugated donor beads (2.5 µL at 40 µg/mL). The mixture was then incubated at room temperature in the dark for 7–28 days. The chemical emissions (Alpha photon counts) were measured using an EnSpire Alpha microplate reader (Revvity), as described previously [19]. We calculated specific reactions by subtracting the emission counts of the GST control from the counts of GST-JMJD6 fusion protein.

Furthermore, we measured the serum levels of PD-L1 using a commercially available enzyme-linked immunosorbent assay (ELISA) kit for PD-L1 (R&D Systems, Minneapolis, MN, USA) [29]. The cutoff value of PD-L1 was 65.62 pg/mL, which produced the lowest *p*-value in the survival analysis.

### 4.6. TCGA Database

The RNA expression levels and survival prognosis data of 144 patients with EC were collected from the TCGA database and downloaded from OncoLnc (accessed on 15 March 2024, http://www.oncolnc.org), as reported previously [55].

### 4.7. Tissue Collection and Immunohistochemistry Staining

The tissue microarray (TMA) was composed of 53 samples from 18 different donors. There were 7 normal tissues, 17 esophageal carcinoma samples, 18 inflammation samples, 8 margin of carcinoma, 1 blood vessel sample, 1 smooth muscle, and 1 hyperplasia sample (diameter: 1.5 mm, thickness: 5 µm) in the TMA construction (CC02-02, Cybrdi, Frederick, MD, USA). Tissues were fixed in 4% buffered formalin and then embedded in paraffin. The TMA manufacturing process has been described previously [56]. Formalin-fixed paraffin-embedded atherosclerosis tissues from patients were cut into 4 µm thick sections. All tissue samples were deparaffinized, blocked with a detection kit (ab64261, Abcam, Cambridge, UK), reacted with primary anti-JMJD6 antibodies (rabbit polyclonal antibodies; GeneTex, Irvine, CA, USA), the anti-vimentin antibody (mouse monoclonal antibody; Agilent, Tokyo, Japan), and the anti-smooth muscle actin antibody (mouse monoclonal antibody; Agilent) at 2 µg/mL at 4 °C overnight, and then incubated with biotinylated anti-rabbit or anti-mouse IgG. The sections were incubated with streptavidin-conjugated horseradish peroxidase reagent. Finally, the reaction was visualized using a diaminobenzidine chromogen, and the sections were counterstained with hematoxylin, dehydrated, and mounted for storage.

### 4.8. Statistical Analysis

Our statistical analysis was based on the standard procedures used in biomedical research. We utilized the Mann–Whitney *U* test and the Kruskal–Wallis test (Mann–Whitney *U* test with Bonferroni correction) to determine significant differences between two groups and between three or more groups, respectively. Correlations were calculated using Spearman’s correlation analysis and logistic regression analysis. All statistical data were analyzed using GraphPad Prism 5 (GraphPad Software, Inc., La Jolla, CA, USA). To assess the predictive values of the putative disease markers, we conducted ROC curve analysis and then set cutoff values to maximize the sums of sensitivity and specificity. Patient survival was evaluated using the Kaplan–Meier method, and survival differences were compared between groups using the log-rank test; the cutoff values were identified using the X-tile software (Version 3.6.1; Yale University, New Haven, CT, USA). All statistical tests were two-tailed, and *p*-values less than 0.05 were considered statistically significant. These analytical methods ensured a reliable and rigorous statistical analysis of the study’s results.

## 5. Conclusions

In conclusion, our study identified s-JMJD6-Abs as a promising biomarker for inflammation-related diseases. Elevated s-JMJD6-Ab levels were observed in patients with conditions such as ischemic stroke, AMI, DM, and various cancers, when compared to HDs. Importantly, higher levels of s-JMJD6-Abs were associated with certain inflammatory markers and correlated with atherosclerosis severity. Notably, positive JMJD6 autoantibody levels can improve postoperative survival for patients with EC. Immunohistochemical analysis confirmed the overexpression of JMJD6 in inflamed mucosa and atherosclerotic plaques. These findings suggest that JMJD6 autoantibodies could serve as diagnostic biomarkers and could have prognostic value for some specific inflammation-related diseases. Future research may explore combining JMJD6 with other biomarkers to improve prognostic accuracy.

## Figures and Tables

**Figure 1 ijms-25-04935-f001:**
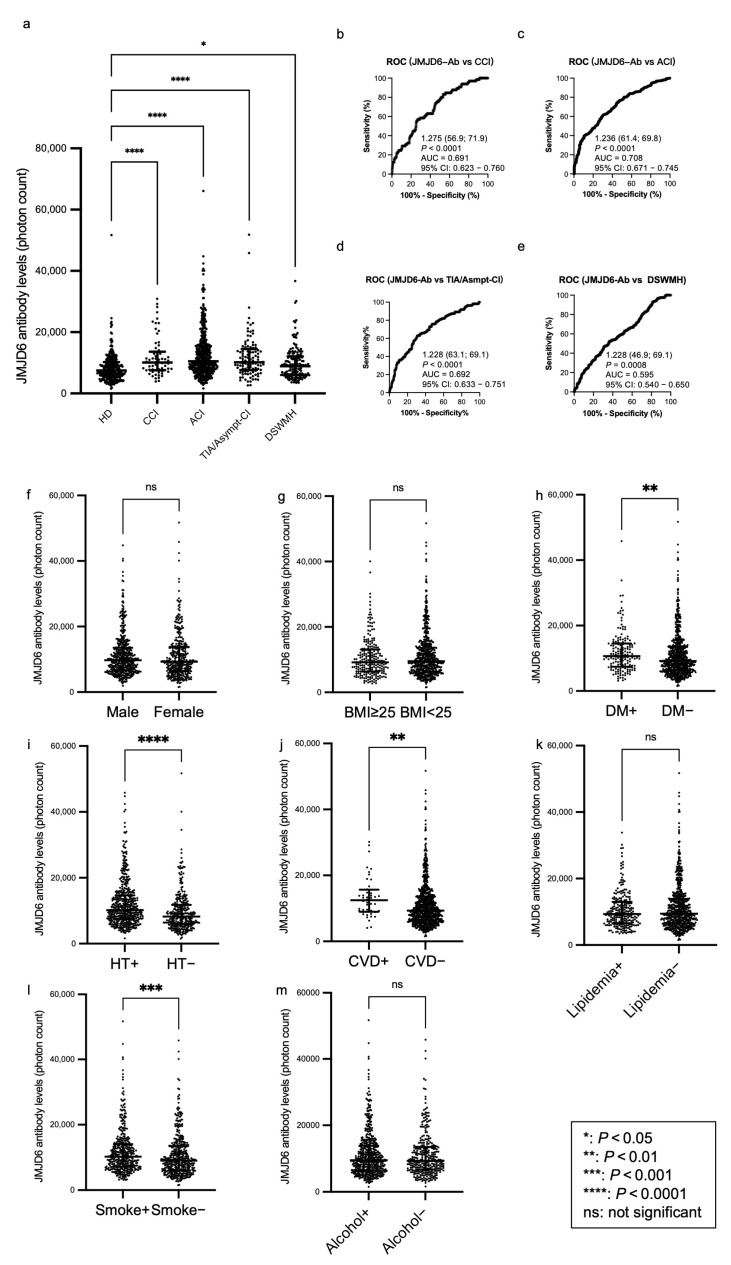
Comparison of the levels of serum anti-JMJD6 antibodies (s-JMJD6-Abs) in healthy donors (HDs) and patients with chronic cerebral infarction (CCI), acute cerebral infarction (ACI), transient ischemic attack (TIA)/asymptomatic cerebral infarction (Asympt-CI), and deep and subcortical white matter hyperintensity (DSWMH). (**a**) The figure shows the scatter dot plot of the levels of s-JMJD6-Abs examined using amplified luminescence proximity homogeneous assay-linked immunosorbent assay (AlphaLISA). The bars represent the medians and quartiles. The *p*-values were calculated using the Kruskal–Wallis test (Mann–Whitney *U* test with Bonferroni correction applied). Table 1 summarizes the exact values. (**b**–**e**) The capacities of s-JMJD6-Abs for detecting CCI (**b**), ACI (**c**), TIA/Asympt-CI (**d**), and DSWMH (**e**) were assessed through receiver operating characteristic (ROC) curve analysis. The numbers in the figures indicate the cutoff values for marker levels, and the numbers in parentheses indicate sensitivity (left) and specificity (right). *p*-values, areas under the curve (AUCs), and 95% confidence intervals (95% CI) are also shown. (**f**–**m**) The associations of s-JMJD6-Ab levels with sex (**f**), body mass index (BMI) (**g**), diabetes mellitus (DM) (**h**), hypertension (HT) (**i**), cardiovascular diseases (CVDs) including acute myocardial infarction and unstable angina pectoris (**j**), lipidemia (**k**), smoking (**l**), and alcohol drinking (**m**) were examined in the Sawara Hospital cohort. The *p*-values were calculated using the Mann–Whitney *U* test, and the bars represent the medians and quartiles; “+” indicates “presence of”; “−” indicates “absence of”.

**Figure 2 ijms-25-04935-f002:**
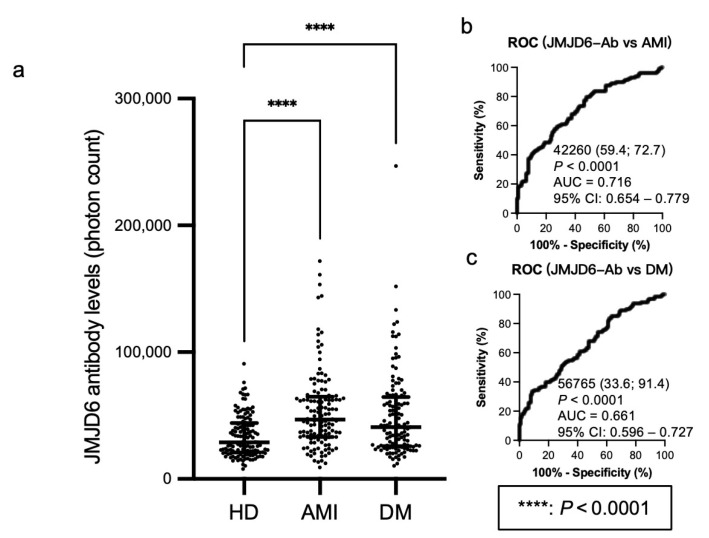
Comparison of the levels of serum anti-JMJD6 antibodies (s-JMJD6-Abs) in healthy donors (HDs) and patients with acute myocardial infarction (AMI) and diabetes mellitus (DM). (**a**) This figure shows the levels of s-JMJD6-Abs examined by amplified luminescence proximity homogeneous assay-linked immunosorbent assay (AlphaLISA). Antibody levels are represented by Alpha photon counts and shown in a scatter dot plot; the horizontal lines represent medians, as well as the 25th and 75th percentiles. The *p*-values were calculated using the Kruskal–Wallis test. (**b**,**c**) The abilities of s-JMJD6-Ab levels to detect AMI (**b**) and DM (**c**) were assessed by receiver operating characteristic (ROC) curve analysis. The numbers in (**b**,**c**) are cutoff values, sensitivity (left), specificity (right), *p*-values, area under the curve (AUC) values, and 95% confidence intervals (95% CI).

**Figure 3 ijms-25-04935-f003:**
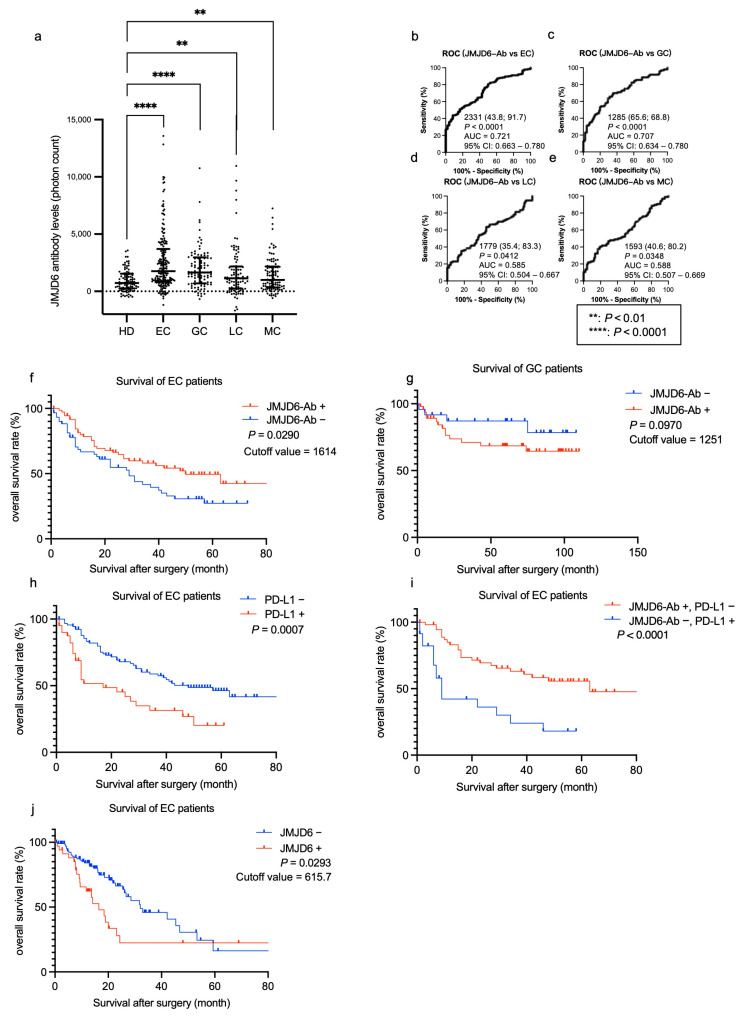
Comparison of the levels of serum anti-JMJD6 antibodies (s-JMJD6-Abs) in healthy donors (HDs) and patients with esophageal cancer (EC), gastric cancer (GC), lung cancer (LC), and mammary cancer (MC). (**a**) The figure shows the levels of s-JMJD6-Abs examined by amplified luminescence proximity homogeneous assay-linked immunosorbent assay (AlphaLISA). Antibody levels are represented by Alpha photon counts and shown in a scatter dot plot; the horizontal lines represent medians and 25th and 75th percentiles. The *p*-values were calculated using the Kruskal–Wallis test. (**b**–**e**) The abilities of s-JMJD6-Ab levels to detect EC (**b**), GC (**c**), LC (**d**), and MC (**e**) were assessed by receiver operating characteristic (ROC) curve analysis. The numbers in the figures are cutoff values for the s-JMJD6-Ab levels, sensitivity (left), specificity (right), the area under the curve (AUC), and 95% confidence interval (95% CI), respectively. (**f**,**g**) The overall survival in patients with EC and GC between the s-JMJD6-Ab-positive (JMJD6-Ab+) and s-JMJD6-Ab-negative (JMJD6-Ab−) groups (*p* = 0.0290, *p* = 0.0970, respectively) was compared and analyzed through a log-rank test. The cutoff values were set at the lowest *p*-values using X-tile, and the *p*-values and cutoff values of s-JMJD6-Abs in patients with EC or GC are shown. The values that were lower than cutoff values were considered to be negative, and those of higher-than-cutoff values were positive values. (**h**,**i**) The overall survival of patients was stratified by their PD-L1 status (PD-L1-positive (PD-L1+) or PD-L1-negative (PD-L1−)) alone (**h**) and in combination with the s-JMJD6-Ab levels (**i**). (**j**) The overall survival status of patients with EC was assessed according to JMJD6 mRNA expression levels (JMJD6-positive group, JMJD6+; or JMJD6-negative group, JMJD6−) collected from the TCGA database. The cutoff value was set at the lowest *p*-value from X-tile, and the cutoff value and *p*-value for JMJD6 mRNA expression levels in patients with EC are shown. Values below the cutoff were considered negative and values above the cutoff were considered positive.

**Figure 4 ijms-25-04935-f004:**
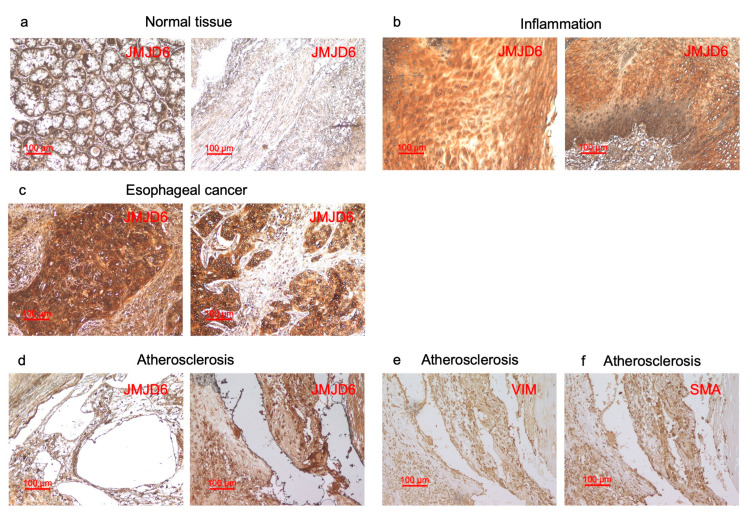
Immunohistochemical staining of tissue samples against JMJD6 antibody. (**a**–**c**) This figure shows that paraffin-embedded sections of normal tissues of esophagus (**a**), inflamed mucosa of esophagus (**b**), and esophageal carcinoma (**c**) were stained with the antibody against human JMJD6. (**d**–**f**) This figure shows that surgically resected atherosclerotic plaques were stained using antibodies against human JMJD6 (**d**), vimentin (VIM) (**e**), and smooth muscle actin (SMA) (**f**). The scale bar indicating 100 µm is shown in red.

**Table 1 ijms-25-04935-t001:** Comparing the serum anti-JMJD6 antibody (s-JMJD6-Ab) levels between healthy donors (HDs) and patients with chronic cerebral infarction (CCI), acute cerebral infarction (ACI), transient ischemic attack/asymptomatic cerebral infarction (TIA/Asympt-CI), and deep and subcortical white matter hyperintensity (DSWMH).

Sample Information	HD	CCI	ACI	TIA/Asympt-CI	DSWMH
Total sample number	285	65	464	111	162
Male/female	188/97	47/18	271/193	68/43	88/74
Age (median ± SD)	53.5 ± 11.7	73.0 ± 9.3	79.0 ± 11.5	73.0 ± 11.9	67.0 ± 10.0
Patient group	Type of value		s-JMJD6-Ab		
HD	Median		7497		
	Cutoff value		16,361		
	Positive no.		14		
	Positive rate (%)		4.9%		
CCI	Median		10,118		
	Positive no.		13		
	Positive rate (%)		**20.0%**		
	*p*-value (vs. HD)		**<0.0001**		
ACI	Median		10486		
	Positive no.		**101**		
	Positive rate (%)		**21.8%**		
	*p*-value (vs. HD)		**<0.0001**		
TIA/Asympt-CI	Median		10150		
	Positive no.		17		
	Positive rate (%)		**15.3%**		
	*p*-value (vs. HD)		**<0.0001**		
DSWMH	Median		8891		
	Positive no.		14		
	Positive rate (%)		8.6%		
	*p*-value (vs. HD)		**0.0025**		

The upper panel of Table 1 provides detailed information of participants, including the total numbers, the numbers of males and females, and their age distributions (median ± SD). The lower panel summarizes the results of the s-JMJD6-Ab with purified JMJD6 protein as an antigen. Cutoff values for positive antibody levels were determined at the 95th percentile value of the HDs. The *p*-values were calculated using the Kruskal–Wallis test, and multiple comparisons were conducted using Bonferroni correction. SD, standard deviation; no., number; vs., versus. *p*-values of <0.05 and positive rates of >10% are marked in bold.

**Table 2 ijms-25-04935-t002:** Spearman’s correlation analysis between the serum anti-JMJD6 antibody levels and clinical features in the Sawara Hospital cohort.

Parameter	Rho Value	95% Confidence Interval	*p*-Value	Number of Pairs
Age	0.3173	0.2532 to 0.3786	**<0.0001**	839
BMI *	−0.0797	−0.1488 to −0.0098	**0.0215**	832
IMT (r)	0.2507	0.1744 to 0.3240	**<0.0001**	640
IMT (l)	0.2661	0.1904 to 0.3386	**<0.0001**	641
max IMT	0.2761	0.2009 to 0.3481	**<0.0001**	642
A/G	−0.1357	−0.2047 to −0.0653	**0.0001**	808
AST	0.0421	−0.0278 to 0.1116	0.2238	836
ALT	−0.0180	−0.0877 to 0.0520	0.6045	835
ALP	0.0989	0.0266 to 0.1701	**0.0059**	775
LDH	0.0541	−0.0169 to 0.1245	0.1243	810
tBil	−0.0469	−0.1171 to 0.0237	0.1801	818
CHE	−0.1352	−0.2129 to −0.0558	**0.0006**	636
γ-GTP	0.0389	−0.0334 to 0.1107	0.2778	782
TP	−0.1581	−0.2264 to −0.0883	**<0.0001**	812
ALB	−0.2019	−0.2686 to −0.1334	**<0.0001**	821
BUN	0.0442	−0.0258 to 0.1137	0.2022	834
Creatinin	0.0213	−0.0488 to 0.0912	0.5399	830
eGFR	−0.0540	−0.1274 to 0.0200	0.1406	746
UA	−0.0093	−0.0909 to 0.0723	0.8176	612
AMY	−0.0430	−0.1313 to 0.0459	0.3288	517
T-CHO	−0.1239	−0.1966 to −0.0498	**0.0008**	733
HDL-c	−0.0222	−0.1087 to 0.0648	0.6072	541
TG	−0.0989	−0.1813 to −0.0152	**0.0173**	579
Na^+^	0.0093	−0.0611 to 0.0797	0.7892	821
K^+^	−0.0573	−0.1272 to 0.01320	0.1009	821
Cl^−^	0.0418	−0.0287 to 0.1119	0.2312	821
CRP	0.1370	0.0555 to 0.2167	**0.0007**	604
WBC	0.0702	0.0003 to 0.1394	**0.0426**	834
Neutrophils	0.0818	0.004362 to 0.1583	**0.0331**	679
Eosinophils	−0.0360	−0.1132 to 0.04157	0.3486	679
Basophils	−0.0566	−0.1335 to 0.02095	0.1406	679
Monocytes	0.0516	−0.02601 to 0.1285	0.1796	679
Lymphocytes	−0.1083	−0.1842 to −0.03106	**0.0047**	679
RBC	−0.1118	−0.1802 to −0.0422	**0.0012**	834
HGB	−0.0830	−0.1520 to −0.0132	**0.0165**	834
HCT	−0.0793	−0.1484 to −0.0095	**0.0220**	834
MCV	0.0879	0.0181 to 0.1568	**0.0111**	834
MCH	0.0781	0.0083 to 0.1472	**0.0241**	834
MCHC	−0.0308	−0.1004 to 0.0392	0.3750	834
RDW	0.0836	0.0138 to 0.1526	**0.0158**	834
PLT	−0.0966	−0.1653 to −0.0269	**0.0052**	834
MPV	−0.0109	−0.0807 to 0.0591	0.7540	834
PCT	−0.0956	−0.1644 to −0.0259	**0.0057**	834
PDW	−0.0470	−0.1165 to 0.0230	0.1751	834
BS	0.1343	0.0622 to 0.2049	**0.0002**	772
HbA1c	0.0571	−0.0225 to 0.1360	0.1474	645
BP	0.0946	0.0102 to 0.1776	**0.0238**	571
Smoking period	0.1515	0.0761 to 0.2253	**<0.0001**	699
Alcohol frequency	0.0449	−0.0319 to 0.1212	0.2382	692

The Sawara hospital cohort includes 285 healthy donors, 65 patients with chronic cerebral infarction, 464 patients with acute cerebral infarction, 92 patients with transient ischemic attack, 19 patients with asymptomatic cerebral infarction, and 162 patients with deep and subcortical white matter hyperintensity. * BMI: body mass index. IMT (r): right-sided intima–media thickness. IMT (l): left-sided intima–media thickness. max IMT: maximum intima–media thickness. A/G: albumin/globulin ratio. AST: aspartate aminotransferase. ALT: alanine aminotransferase. ALP: alkaline phosphatase. LDH: lactate dehydrogenase. tBil: total bilirubin. CHE: choline esterase. γ-GTP: gamma-glutamyl transpeptidase. TP: total protein. ALB: albumin. BUN: blood urea nitrogen. eGFR: estimated glomerular filtration rate. UA: uric acid. AMY: amylase. T-CHO: total cholesterol. HDL-c: high-density lipoprotein cholesterol. TG: triglyceride. CRP: C-reactive protein. WBC: white blood cell count. RBC: red blood cell count. HCT: hematocrit. MCV: mean corpuscular volume. MCH: mean corpuscular hemoglobin. MCHC: mean corpuscular hemoglobin concentration. RDW: red cell distribution width. PLT: platelet count. MPV: mean platelet volume. PCT: procalcitonin. PDW: platelet distribution width. BS: blood glucose. HbA1c: glycated hemoglobin. BP: blood pressure. *p*-values of <0.05 are marked in bold.

**Table 3 ijms-25-04935-t003:** Logistic regression analysis of the predictors of acute cerebral infarction.

	Univariate Analysis			Multivariate Analysis		
	Odds Ratio	95% CI	*p*-Value	Odds Ratio	95% CI	*p*-Value
Sex (Male)	0.78	0.54 to 1.12	0.1700			
Age > 65	25.30	15.70. to 40.90	**<0.0001**	15.7	9.00 to 27.40	**<0.0001**
DM	5.98	3.02 to 11.80	**<0.0001**	4.75	2.04 to 11.10	**0.0003**
HT	9.72	6.43 to 14.70	**<0.0001**	5.16	3.02 to 8.81	**<0.0001**
CVDs	9.14	2.16 to 38.60	**0.0026**	3.17	0.67 to 15.00	0.1470
Lipidemia	1.32	0.86 to 2.01	0.2020			
Obesity (BMI ≥ 25)	0.95	0.64 to 1.41	0.7990			
Smoking	1.38	0.96 to 1.99	0.0781			
s-JMJD6-Ab	3.80	2.60 to 5.57	**<0.0001**	3.77	2.19 to 6.50	**<0.0001**

The anti-serum JMJD6 antibody (s-JMJD6-Ab) cutoff value of 9274 for the ACI group was applied in the univariate analysis. Univariate data with *p*-values of <0.05 were included in the multivariate analysis and are marked in bold. DM: diabetes mellitus. HT: hypertension. CVDs: cardiovascular diseases. BMI: body mass index. 95% CI: 95% confidence interval.

**Table 4 ijms-25-04935-t004:** Comparing the serum anti-JMJD6 antibody (s-JMJD6-Ab) levels between healthy donors (HD) and patients with acute myocardial infarction (AMI) or diabetes mellitus (DM).

Sample Information	HD	AMI	DM
Total sample number	128	128	128
Male/female	72/56	105/23	72/56
Age (median ± SD)	57.0 ± 5.6	61.0 ± 8.5	60.0 ± 9.2
Patient group	Type of value	s-JMJD6-Ab	
HD	Median	28,808	
	Cutoff value	66,732	
	Positive no.	6	
	Positive rate (%)	4.7%	
AMI	Median	46,884	
	Positive no.	28	
	Positive rate (%)	**21.9%**	
	*p*-value (vs. HD)	**<0.0001**	
DM	Median	50,092	
	Positive no.	27	
	Positive rate (%)	**21.1%**	
	*p*-value (vs. HD)	**<0.0001**	

The upper panel provides details of participants, including the total numbers of collected samples, and the numbers of samples collected from male and female participants and their age distributions (median ± SD). The lower panel summarizes the results of s-JMJD6-Abs with purified JMJD6 protein. Positive antibody levels were determined using cutoff values set at the 95th percentile value of the HDs. The Kruskal–Wallis test and Bonferroni correction were employed for *p*-value calculation and multiple comparisons. SD, standard deviation. no., number. vs., versus. *p*-values of <0.05 and positive rate of >10% are marked in bold.

**Table 5 ijms-25-04935-t005:** Comparing the serum anti-JMJD6 antibody (s-JMJD6-Ab) levels between healthy donors (HDs) and patients with esophageal cancer (EC), gastric cancer (GC), lung cancer (LC), and mammary cancer (MC).

Sample Information	HD	EC	GC	LC	MC
Total sample number	96	192	96	96	96
male/female	51/45	155/37	68/28	42/54	58/38
Age (median ± SD)	57.0 ± 6.0	68.0 ± 9.8	69.0 ± 10.6	63.0 ± 13.3	69.0 ± 9.6
Subject	Type of value		s-JMJD6-Ab		
HD	Median		734		
	Cutoff value		2629		
	Positive no.		4		
	Positive rate (%)		4.2%		
EC	Median		1755		
	Positive no.		67		
	Positive rate (%)		**34.9%**		
	*p*-value (vs. HD)		**<0.0001**		
GC	Median		1646		
	Positive no.		29		
	Positive rate (%)		**30.2%**		
	*p*-value (vs. HD)		**<0.0001**		
LC	Median		1140		
	Positive no.		21		
	Positive rate (%)		**21.9%**		
	*p*-value (vs. HD)		**0.0031**		
MC	Median		992		
	Positive no.		19		
	Positive rate (%)		**19.8%**		
	*p*-value (vs. HD)		**0.0054**		

The upper panel of Table 5 provides information on the participants, including the total numbers of samples collected, the numbers of samples obtained from male and female participants, and their age distributions (median ± SD). The lower panel shows the results of the s-JMJD6-Abs with purified JMJD6 protein as an antigen. Positive antibody levels were determined according to the cutoff values set at the 95th percentile of the HDs. The Kruskal–Wallis test and Bonferroni correction were used for *p*-value calculation and multiple comparisons. SD, standard deviation. no., number. vs., versus. *p*-values of <0.05 and positive rate of >10% are marked in bold.

## Data Availability

The data sets used and/or analyzed during the current study are available from the corresponding author on reasonable request.

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
