# Peer review of "JMJD6 Autoantibodies as a Potential Biomarker for Inflammation-Related Diseases"

_ijms, 2024, doi:10.3390/ijms25094935_

Round 1
Reviewer 1 Report (Previous Reviewer 1)
Comments and Suggestions for Authors
The manuscript focuses now on inflammation and anti-JMJD6 auto-antibodies. The mixture of several diseases persists but with a more logical common feature for all of them. Some imprecisions and the lack of data reproducibility remain as the most important the concern of this manuscript.
11. Units of anti-JMJD6 auto-antibodies levels are still lacking; even if they are arbitrary units, they should be indicated.
22. It seems that crude values for antibodies levels obtained directly from the measurement instrument vary from run to run, as also happens for other methods, i.e. spectrophotometry or chromatography. The use of calibration standards is the solution to have comparable results from run to run, without the extreme variations as observed in this study. Why didn’t author included standard samples of a known autoantibody level to obtain comparable results for all the samples?
33. Authors did not indicate whether ALL samples from the Sawara Hospital and their corresponding healthy controls were run at the same time in order to avoid the inter-assay variations. The same observation applies for samples obtained from patients with AMI, DM, or cancer, and their corresponding controls. As values of antibodies levels from one day to another vary as much as 30 times, the only way to obtain comparable data among groups is to process all the samples at the same time. Was it the case in this study? Indicate these details in methods section.
44. Table 2 does not apport anything to the main idea of the manuscript; it could be deleted or presented as supplemental material. By the way, height and weight are redundant with BMI.
55. The use of 3 different cutoff values is confusing and suggest a bias of the statistical analysis; for tables 1 ,4 and 5, the cutoff values were the average of the HDs’ value plus two SDs. For logistic regression, the cutoff values were those derived from ROC curves, and for the analyses of cancer patients’ survival, cutoff values obtained from X-tile software. The latter is the most confusing and it is unclear whether values of antibodies below this threshold were considered as negative to autoantibodies.
66. The use of median and mean for the same data is not correct. If data is not normally distributed, then the median is the best to indicate the central tendency. If data present a normal distribution, the median+/- SD is the best expression. Perform an analysis of data distribution and use the most appropriate expression for the central tendency values.
77. If PDL-1 plasma levels is a good biomarker of cancer outcomes, a comparison, i.e. multifactorial correlation analysis, between PDL-1 and anti-JMJD6 auto-antibodies would be of interest.
Comments on the Quality of English LanguageEnglish is correct.
Author Response
Please see attached file.

Reviewer 2 Report (New Reviewer)
Comments and Suggestions for Authors
The manuscript entitled JMJD6 Autoantibodies as a Potential Biomarker for Inflammation-related Diseases brings a new molecule as an important inflammation biomarker in several diseases (myocardial infarction, stroke, diabetes mellitus and cancer)
Observations
Abstract
- line 43 - hypertension is not and inflammatory indicator (only CRP is an inflammation indicator, the others are diseases associated with inflammation).
Introduction
- line 78 - replace sera with serum ; please replace this word everywhere in your manuscript.
Please specify at the end of Introduction, as the aim of this study, why nis important to identify serum anti-JMJD6-Abs as biomarker for inflammation associated with myocardial infraction, stroke, DM or cancer since we already know these diseases have inflammatory reaction associated.
- line 191 - what do you mean by asymptomatic cerebral infarction?
- line 336 - indicate references
- line 353 - indicate references
Conclusions
Please write the Conclusions more detailed, refer to your results, their importance in clinical practice and what is new in your study.
Author Response
Please see attached file.

Reviewer 3 Report (New Reviewer)
Comments and Suggestions for Authors
Dear Authors,
I think your manuscript talks about a topic of great interest with potential implications for clinical practice.
I ask you to answer the following questions:
1) INTRODUCTION: The presence of JMJD6 in addition to the situations in which you have described it, can Autoantibodies also manifest itself in more commonly encountered inflammatory conditions (e.g. flu, etc.)?
2) Figure 1 is blurry and the definition of the image should be increased
Overall, the data described are supported by an excellent discussion and the conclusion schematically summarizes the results.
The bibliography is cited correctly and is relevant to the contents.
Round 2
Reviewer 1 Report (Previous Reviewer 1)
Comments and Suggestions for Authors
Authors did not look for an internal standard for their method, so the variability of the reported vaules remains of about 30 times for the same kind of samples; from my point of view, this is unacceptable because data is not reproducible, and conclusions lacks of a solid methodological support.
This manuscript is a resubmission of an earlier submission. The following is a list of the peer review reports and author responses from that submission.
Round 1
Reviewer 1 Report
Comments and Suggestions for Authors
In this manuscript, Zhang et al. determined the presence of autoantibodies against the protein JMJD6 in chronic cerebral infarction (CCI), acute cerebral infarction (ACI), transient ischemic attack (TIA), asymptomatic cerebral infraction (asympt-CI), deep and subcortical white matter hyperintensity (DSWMH), acute myocardial infarction (AMI), diabetes mellitus (DM), esophageal cancer (EC), gastric cancer (GC), lung cancer (LC), and mammary cancer (MC). The main results include the increased levels of autoantibodies compared to the corresponding control group of each subanalysis.
The topic is of relevance, but the manuscript is too extensive, difficult to follow in some sections, and there are important issues that call into question the validity of the results.
1. My main concern are the autoantibodies levels in “healthy donors”. First, Authors statement indicates that “serum samples from HDs were obtained from patients without abnormalities in cranial magnetic resonance imaging”; are control HD sera the same for the 3 different subgroups, i.e. stroke related diseases, AMI/DM, and different cancers? It is imperative to detail the main characteristics of control groups. Second, even if there were 3 different control groups for the subanalyses, the autoantibody levels are strikingly discrepant, up to 34 times for the same kind of individuals; in table 1, HD values are 8359±4741, in table 3, HD values are 33844±16513, and in table 4 the autoantibody plasma levels for HD are 930±932. This discrepancy indicates that the method for JMJD6 autoantibody determination lacks accuracy and repeatability, and it is impossible to obtain valid conclusions with such method. Also, units for autoantibody plasma levels were omitted.
2. It is unclear whether AlphaLISA is an in-house test. If so, quality controls should be included in the manuscript.
3. The inclusion of 11 so different clinical entities is not justified; atherosclerosis is not related to all these diseases. Moreover, an analyte that increases in such different clinical circumstances is an irrelevant and usefulness analyte for diagnostics and risk evaluation of a particular disease.
4. Which is the physiological role of JMJD6? Authors indicate 3 different activities including methylase (of histones?), hydroxylase (any protein?), and phosphatidylserine receptor. I was unable to verify the two latter activities and Authors did not provide adequate bibliographic support to their statements. Therefore, data interpretation is not acceptable.
5. Statistical analysis is superficial. There are important differences between controls HD and patients, and autoantibodies correlate with several other parameters. Consequently, statistical analyses should be corrected by potential confounding variables such age, other co-morbidities, drug intake, blood pressure, etc. Also, data distribution was not assessed.
6. Patient selection should be detailed since it is not clear for example whether some AMI patients are also diabetic.
7. Lines 196-197 indicate that dots represent outliers, but the figure is a dots graph.
8. The level of significance asterisks, i.e. *, **, etc., is not indicated in the figures, even if it is mentioned in the legends.
9. How was documented the survival of cancer patients? Was it a follow-up study?
Comments on the Quality of English Language
English is acceptable.
Author Response
Review Report Form
Reviewer 1
Comments and Suggestions for Authors
In this manuscript, Zhang et al. determined the presence of autoantibodies against the protein JMJD6 in chronic cerebral infarction (CCI), acute cerebral infarction (ACI), transient ischemic attack (TIA), asymptomatic cerebral infraction (asympt-CI), deep and subcortical white matter hyperintensity (DSWMH), acute myocardial infarction (AMI), diabetes mellitus (DM), esophageal cancer (EC), gastric cancer (GC), lung cancer (LC), and mammary cancer (MC). The main results include the increased levels of autoantibodies compared to the corresponding control group of each subanalysis.
The topic is of relevance, but the manuscript is too extensive, difficult to follow in some sections, and there are important issues that call into question the validity of the results.
- My main concern are the autoantibodies levels in “healthy donors”. First, Authors statement indicates that “serum samples from HDs were obtained from patients without abnormalities in cranial magnetic resonance imaging”; are control HD sera the same for the 3 different subgroups, i.e. stroke related diseases, AMI/DM, and different cancers? It is imperative to detail the main characteristics of control groups. Second, even if there were 3 different control groups for the subanalyses, the autoantibody levels are strikingly discrepant, up to 34 times for the same kind of individuals; in table 1, HD values are 8359±4741, in table 3, HD values are 33844±16513, and in table 4 the autoantibody plasma levels for HD are 930±932. This discrepancy indicates that the method for JMJD6 autoantibody determination lacks accuracy and repeatability, and it is impossible to obtain valid conclusions with such method. Also, units for autoantibody plasma levels were omitted.
Because each AlphaLISA reading may yield different results due to factors such as detection temperature, light intensity, oxygen/atmosphere pressure. It is absolutely necessary to obtain the results of plating and measuring at the same time. Only results which were obtained at the same time were compared. It is understandable if the same serum samples yielded different Alpha counts in different experiments.
- It is unclear whether AlphaLISA is an in-house test. If so, quality controls should be included in the manuscript.
AlphaLISA is not an in-house test. It is a commercial technology developed by PerkinElmer, a global leader in life sciences and diagnostics. This technology is widely used in various research areas, including drug discovery, biomarker analysis, and diagnostics, due to its sensitivity, versatility, and suitability for high-throughput applications.
- The inclusion of 11 so different clinical entities is not justified; atherosclerosis is not related to all these diseases. Moreover, an analyte that increases in such different clinical circumstances is an irrelevant and usefulness analyte for diagnostics and risk evaluation of a particular disease.
As you indicated, atherosclerosis is not a sole cause for cerebro- and cardiovascular disease, yet it is one of the major causes for them. Its related diseases include DM, ischemic stroke, and AMI, all of which are multifactorial diseases. Thus, for the research on atherosclerosis, we think it necessary to examine as many related diseases as possible using as many samples as possible. By adopting the reproducible and universal results, we can get closer to the truth.
- Which is the physiological role of JMJD6? Authors indicate 3 different activities including methylase (of histones?), hydroxylase (any protein?), and phosphatidylserine receptor. I was unable to verify the two latter activities and Authors did not provide adequate bibliographic support to their statements. Therefore, data interpretation is not acceptable.
Thanks for your suggestion. Based on the suggestion, we modified our paper.
We changed Introduction as follows.
JMJD6 functions as a protein demethylase towards histone substrates, a hydroxylase targeting non-histone proteins, and a phosphatidylserine receptor, influencing RNA splicing, lipid metabolism, and apoptosis [24,26–29].
We changed Discussion section as below.
By contrast, JMJD6 is a phosphatidylserine receptor, and a monoclonal anti-JMJD6 antibody inhibited the phagocytosis of phosphatidylserine-expressing apoptotic cells [29,44]. There are some studies that knockouts of the JMJD6 gene result to a larger number of non-phagocytosed apoptotic cells compared to wild-type animals [45,46], and a monoclonal anti-JMJD6 antibody inhibited the phagocytosis of phosphatidylserine-expressing apoptotic cells [29,44]. Thus, a similar phenomenon could also be caused by s-JMJD6-Abs.
- Statistical analysis is superficial. There are important differences between controls HD and patients, and autoantibodies correlate with several other parameters. Consequently, statistical analyses should be corrected by potential confounding variables such age, other co-morbidities, drug intake, blood pressure, etc. Also, data distribution was not assessed.
It is important to consider on the confounding variables. To examine such variables, we performed logistic regression analysis and added them in results 2.3. The results showed that s-JMJD6-Ab level was an independent predictor of ACI.
We revised Results as follows.
Using the clinical risk factors and cutoff values obtained earlier, logistic regression analysis in the ACI group revealed that not only clinical risk factors such as age > 65 years (P < 0.0001), DM (P < 0.0001), HT (P < 0.0001), and CVD (P = 0.0026) but also high antibody level (P < 0.0001) were associated with ACI (Table 3). Multivariate analysis of the univariate data with P values of < 0.05 revealed that in addition to the atherosclerotic factors mentioned earlier, elevated s-JMJD6-Ab level was an independent predictor of ACI (odds ratio: 3.77, 95% CI: 2.19–6.50, P < 0.0001; Table 3). These results suggested that s-JMJD6-Ab levels may serve as an independent biomarker for ACI and help improve the prediction and diagnosis of ACI risk.
And Discussion was refined as follows.
According to the multivariate analysis, s-JMJD6-Abs emerged as an independent predictor in ACI (Table 3). These findings underscore the association between s-JMJD6-Abs and the development of atherosclerosis, culminating in the onset of ACI.
- Patient selection should be detailed since it is not clear for example whether some AMI patients are also diabetic.
We appreciate the reviewer's attention to detail regarding patient selection criteria. We apologize for confusion regarding the inclusion of diabetic status among patients with acute myocardial infarction (AMI). Upon reviewing our data and patient records, we confirm that information of the diabetic status of AMI patients.
Firstly, although a portion of patients with diabetes mellitus (DM) have a history of cardiovascular diseases, the number within this subgroup is extremely limited and negligible in its impact on the results. Secondly, while we do not possess records of diabetic status specifically for AMI patients, it is worth noting that serum samples from AMI patients were collected during the acute phase of the AMI event. Therefore, these samples can be regarded as indicative of the acute phase of AMI without consideration of diabetic status. Furthermore, it is noteworthy that there is an overlap between the populations of patients with AMI or DM in real-world scenarios. Consequently, the exploration of diabetic status among AMI patients becomes a part of the etiological analysis of AMI, which is important but not the core of our research.
- Lines 196-197 indicate that dots represent outliers, but the figure is a dots graph.
Thanks for your indication. 'dots represent outliers' was a mistake, and we deleted the words. We have changed the legends of Figure 2 as below.
(a) The figure shows the levels of s-JMJD6-Abs examined by amplified luminescence proximity homogeneous assay-linked immunosorbent assay (AlphaLISA). Antibody levels are represented by Alpha photon counts and shown in a scatter dot plot; the horizontal lines represent medians. The P-values were calculated using the Kruskal–Wallis test.
- The level of significance asterisks, i.e. *, **, etc., is not indicated in the figures, even if it is mentioned in the legends.
Thanks for your comments. The asterisks of the statistical significance have happened to disappear unexpectedly during the file conversion. We refined the Figure 1–3 with the significance asterisks.
- How was documented the survival of cancer patients? Was it a follow-up study?
The survival analysis is a retrospective study. All the survival of patients were documented before performing the antibody levels, as shown in section 4.1.
The Department of Surgery in Toho University Hospital collected sera from patients with EC, GC, LC, and MC preoperatively between June 2010 and February 2016, and patients with EC or GC were retrospectively followed up until July 2018 or death.

Reviewer 2 Report
Comments and Suggestions for Authors
In this paper, authors examined the potential of using serum JMJD6 autoantibodies as a biomarker for atherosclerosis related diseases. This is an potentially interesting and important finding. However, more evidence will be needed to support the conclusions.
Major concerns:
1, Atherosclerosis is closely associated with hyperlipidemia and hypertension. Authors should also examine the correlations between serum JMJD6 autoantibodies and blood lipid / pressure levels.
2, There are many other biomarker now available for atherosclerosis, such as serum IL1b levels. Authors should at least compare the correlation coefficient of serum JMJD6 autoantibodies with one well-established marker.
Minor:
1, The statistical significance of some panels were not labeled.
2, A moderate language polish will be needed.
Comments on the Quality of English LanguageModerate polish will be needed.
Author Response
Review Report Form
Reviewer 2
Major concerns:
1, Atherosclerosis is closely associated with hyperlipidemia and hypertension. Authors should also examine the correlations between serum JMJD6 autoantibodies and blood lipid / pressure levels.
Thanks for kind suggestions. We examined the lipidemia and hypertension in patients with AIS (Figure 1j,i). It shows that JMJD6-Ab is associated with blood pressure, but not with lipidemia. And we added the logistic regression analysis of these risk factors in ACI (Table 3).
The results were revised as follows.
Using the clinical risk factors and cutoff values obtained earlier, logistic regression analysis in the ACI group revealed that not only clinical risk factors such as age > 65 years (P < 0.0001), DM (P < 0.0001), HT (P < 0.0001), and CVD (P = 0.0026) but also high antibody level (P < 0.0001) were associated with ACI (Table 3). Multivariate analysis of the univariate data with P values of < 0.05 revealed that in addition to the atherosclerotic factors mentioned earlier, elevated s-JMJD6-Ab level was an independent predictor of ACI (odds ratio: 3.77, 95% CI: 2.19–6.50, P < 0.0001; Table 3). These results suggested that s-JMJD6-Ab levels may serve as an independent biomarker for ACI and help improve the prediction and diagnosis of ACI risk.
2, There are many other biomarker now available for atherosclerosis, such as serum IL1b levels. Authors should at least compare the correlation coefficient of serum JMJD6 autoantibodies with one well-established marker.
Thank you for mentioning the other biomarkers. There are already some other biomarkers for atherosclerosis. And we took interleukin-1β, interleukin-6, homocysteine as examples to discuss as below.
In addition, there are already some biomarkers in atherosclerosis, such as serum interleukin-1β, interleukin-6, homocysteine levels [48]. The mechanisms of JMJD6 and other biomarkers on atherosclerosis are still different. Therefore, further studies are necessary to elucidate the intrinsic connections between different biomarkers and the broader effects of JMJD6 autoantibodies in different pathological conditions.
Minor:
1, The statistical significance of some panels were not labeled.
Thanks for your indication. The asterisks of the statistical significance have happened to disappear unexpectedly during the file conversion. We refined the Figure 1–3 with the significance asterisks.
2, A moderate language polish will be needed.
We have polished the English language writing carefully.

Reviewer 3 Report
Comments and Suggestions for Authors
This is an interesting study which investigated the role of serum anti-JMJD6 antibodies (s-JMJD6-Abs) in diagnosing and prognosticating various cardiovascular and cardiovascular-related diseases (e.g. cancer). The methods and well described, and the results are convincing. The Discussion is missing some elements, and the figures require revisions, as detailed below.
1. Figure 1 (a, i-k, m): While the brackets in Figure 1 (g, h, l, and n) are clearly labelled “ns,” the remaining brackets appear to be missing asterisks indicating level of significance (e.g. as demonstrated by Park et al. in Figure 7 [ref]), which appears to be the intention as indicated by the figure legend. The same issue regarding missing asterisks is present in Figure 2 (a) and Figure 3 (a).
ref: Park JH, Lee DK, Kang H, Kim JH, Nahm FS, Ahn E, In J, Kwak SG, Lim CY. The principles of presenting statistical results using figures. Korean J Anesthesiol. 2022 Apr;75(2):139-150. doi: 10.4097/kja.21508. Epub 2022 Mar 3. PMID: 35016496; PMCID: PMC8980283.
2. The AUCs (differentiating between healthy subjects and patients with evidence of disease) reported in this study are quite low. Therefore, claims such as “s-JMJD6-Abs can be used to predict patients with a high early-stage cerebral infarction risk” (line 268-269) are tenuous. The authors refer to the statistically significant difference in mean antibody expression level between healthy subjects and patients at risk for stroke to support their claim, but because no cutoff value results in a test with high sensitivity and specificity (as shown in Figure 1b-f), the actual utility is questionable. This should be recognized in the Discussion, and statements such as the one above should be softened or qualified.
3. The Discussion section must discuss the implications of the findings in context of existing research. Specifically, results from previous studies examining the association between s-JMJD6-Abs and other diseases should be mentioned. Limitations of the current study must also be mentioned.
Author Response
Review Report Form
Reviewer 3
Comments and Suggestions for Authors
This is an interesting study which investigated the role of serum anti-JMJD6 antibodies (s-JMJD6-Abs) in diagnosing and prognosticating various cardiovascular and cardiovascular-related diseases (e.g. cancer). The methods and well described, and the results are convincing. The Discussion is missing some elements, and the figures require revisions, as detailed below.
- 1. Figure 1 (a, i-k, m): While the brackets in Figure 1 (g, h, l, and n) are clearly labelled “ns,” the remaining brackets appear to be missing asterisks indicating level of significance (e.g. as demonstrated by Park et al. in Figure 7 [ref]), which appears to be the intention as indicated by the figure legend. The same issue regarding missing asterisks is present in Figure 2 (a) and Figure 3 (a).
ref: Park JH, Lee DK, Kang H, Kim JH, Nahm FS, Ahn E, In J, Kwak SG, Lim CY. The principles of presenting statistical results using figures. Korean J Anesthesiol. 2022 Apr;75(2):139-150. doi: 10.4097/kja.21508. Epub 2022 Mar 3. PMID: 35016496; PMCID: PMC8980283.
Thanks for your comments. The asterisks of the statistical significance have happened to disappear unexpectedly during the file conversion. We refined the Figure 1–3 with the significance asterisks.
- The AUCs (differentiating between healthy subjects and patients with evidence of disease) reported in this study are quite low. Therefore, claims such as “s-JMJD6-Abs can be used to predict patients with a high early-stage cerebral infarction risk” (line 268-269) are tenuous. The authors refer to the statistically significant difference in mean antibody expression level between healthy subjects and patients at risk for stroke to support their claim, but because no cutoff value results in a test with high sensitivity and specificity (as shown in Figure 1b-f), the actual utility is questionable. This should be recognized in the Discussion, and statements such as the one above should be softened or qualified.
Thank you for bringing this important issue to our attention. Indeed, considering that stroke and AMI are multifactorial diseases, AUC value of 0.7 is not necessarily low (Figure 1 b-f). Different markers may correspond to different causes. We would like to identify and validate as many markers as possible. By combining such novel markers as well as existing markers, more reliable diagnosis and prediction with higher AUC values can be achieved.
- The Discussion section must discuss the implications of the findings in context of existing research. Specifically, results from previous studies examining the association between s-JMJD6-Abs and other diseases should be mentioned. Limitations of the current study must also be mentioned.
Thanks for your suggestion. We think it is necessary to discuss about the current status and future of JMJD6-Ab and other biomarkers of atherosclerosis. So we add the following explanation in Discussion.
Previous research suggested the association between JMJD6 and many diseases, es-pecially s-JMJD6-Ab and colorectal cancer. Here, our study identifies a potential associa-tion between s-JMJD6-Abs and various atherosclerosis-related diseases, and suggests JMJD6-Abs as a biomarker for atherosclerosis-related diseases. However, we acknowledge the limitations regarding the predictive performance of s-JMJD6-Abs alone, as indicated by the modest AUC values. Because atherosclerosis is a multifactorial disease, there may be different markers depending on the causes. In addition, there are already some biomarkers in atherosclerosis, such as serum interleukin-1β, interleukin-6, homocysteine levels [48]. The mechanisms of JMJD6 and other biomarkers on atherosclerosis are still different. Therefore, further studies are necessary to elucidate the intrinsic connections between different biomarkers and the broader effects of JMJD6 autoantibodies in different pathological conditions.

Round 2
Reviewer 2 Report
Comments and Suggestions for Authors
The authors handled all my concerns.
Reviewer 3 Report
Comments and Suggestions for Authors
The previously raised concerns have been adequately addressed. No further comments are necessary.